# Incidence of Concomitant Neoplastic Diseases, Tumor Characteristics, and the Survival of Patients with Lung Adenocarcinoma or Squamous Cell Lung Carcinoma in Tobacco Smokers and Non-Smokers—10-Year Retrospective Single-Centre Cohort Study

**DOI:** 10.3390/cancers15061896

**Published:** 2023-03-22

**Authors:** Błażej Ochman, Paweł Kiczmer, Paweł Ziora, Mateusz Rydel, Maciej Borowiecki, Damian Czyżewski, Bogna Drozdzowska

**Affiliations:** 1Students’ Scientific Society, Department of Pathomorphology, Faculty of Medical Sciences in Zabrze, Medical University of Silesia, 40-055 Katowice, Poland; s71690@365.sum.edu.pl; 2Department of Pathomorphology, Faculty of Medical Sciences in Zabrze, Medical University of Silesia, 40-055 Katowice, Poland; pkiczmer@sum.edu.pl (P.K.); pziora@sum.edu.pl (P.Z.); bdrozdzowska@sum.edu.pl (B.D.); 3Department of Thoracic Surgery, Faculty of Medical Sciences in Zabrze, Medical University of Silesia, 40-055 Katowice, Poland; mrydel@sum.edu.pl (M.R.); dczyzewski@sum.edu.pl (D.C.)

**Keywords:** non-small cell lung carcinoma, non-smoker, lung carcinoma survival

## Abstract

**Simple Summary:**

Lung cancer is one of the biggest public health issues due to its high prevalence and mortality. Currently, increasing consideration is given to the incidence of lung cancer in the individuals with no lifetime history of tobacco smoking. However, up-to-date data on the characteristics of a group of non-smokers with lung cancer are limited. The current literature also contains gaps in the differences in the clinical course of lung cancer in smokers and non-smokers. This study aimed to investigate the differences in tumor characteristics, survival rates, and comorbidities between a group of smokers and a group of non-smokers with lung cancer. The presented results may be used in clinical practice and in shaping future lung cancer prevention programs.

**Abstract:**

Changes in smoking trends and changes in lifestyle, together with worldwide data regarding the incidence of lung cancer in the group of patients with no previous history of smoking, leads to consideration of the differences in the course of the disease, the time of cancer diagnosis, the survival rate, and the occurrence of comorbidities in this group of patients. This study aimed to determine the occurrence of non-smokers among patients undergoing anatomical resection of the lung tissue due to lung carcinoma and to investigate the differences between the course of lung cancer, survival, and the comorbidities in the groups of patients with lung cancer depending on the history of tobacco smoking. The study included a cohort of 923 patients who underwent radical anatomical resection of the lung tissue with lung primary adenocarcinoma or squamous cell carcinoma. The Chi2 Pearson’s test, the *t*-test, the Mann–Whitney U test, the Kaplan–Meier method, the Log-rank test with Mantel correction, and the Cox proportional hazard model were used for data analysis. We observed a significantly higher mean age of smoking patients compared to the mean age of non-smoking patients. The coexistence of former neoplastic diseases was significantly more frequent in the group of non-smokers compared to the group of smoking patients. We did not observe differences depending on smoking status in the tumor stage, grade, vascular and pleural involvement status in the diagnostic reports. We did not observe differences in the survival between smokers vs. non-smokers, however, we revealed better survival in the non-smoker women group compared to the non-smoker men group. In conclusion, 22.11% of the patients undergoing radical anatomical resection of the lung tissue due to lung cancers were non-smokers. More research on survival depending on genetic differences and postoperative treatment between smokers and non-smokers is necessary.

## 1. Introduction

For decades, lung cancer has maintained its position as one of the cancers with the highest incidence rate, furthermore, it is the leading cause of cancer mortality globally. According to GLOBOCAN 2020 data, the incidence of lung cancer was 11.4% with a mortality rate of 18% of all cancer deaths for both sexes worldwide. Differences in incidences and mortality have been observed between women and men: incidence rates were 8.4% and 14.3%, and cancer-related mortality was 13.7% and 21.5% for women and men, respectively [1]. Continuous exposure to tobacco smoke is largely to blame as the primary and predominant cause of lung cancer [2,3]. However, among lung cancer patients there is also a group of patients who have never smoked tobacco, amounting to about 10–25% depending on the examined population, geographic distribution, and methodology of the studies. Due to the high incidence of lung cancer, even a small proportion of patients with no history of smoking may represent a large group of patients [4,5]. Considering the generally decreasing prevalence of smoking worldwide [6,7,8], lung cancer risk factors other than smoking itself and lung cancer non-smoking patients’ characteristics should be under the spotlight, both now and in the future. The relationship between the increased incidence of lung cancer in non-smokers with a family history of cancer, as observed in many studies, seems to support the considered significant role of the influence of genetic factors on the incidence of lung cancer in non-smokers [9,10]. Independent risk factors for the development of lung cancer in non-smokers, in addition to environmental air pollution [11,12] and passive smoking [13], also include previous respiratory diseases, such as asthma, pneumonia, tuberculosis, chronic bronchitis, and viral infections [14,15]. Besides considered genetic factors, obesity and diet may also be especially important in the development of lung adenocarcinoma [16,17]. That may seem even more important due to the higher incidence of adenocarcinoma in comparison to the other histological subtypes of lung cancer in the group of never-smokers [18]. In our study, we analyzed the data of patients operated on for lung cancer in 2012–2021. The analyzed data included the smoking history, gender, concomitant neoplastic diseases other than lung cancer, comorbidities, family cancer history, tumor stage, grade, histologic type, lymphovascular, and pleural invasion, and the patients’ survival data. Currently, the characteristics of non-smokers with lung cancer, including the survival, course differences, tumor stage, and grade at diagnosis are both heterogeneous and limited. Due to the lack of data on smoking in most cancer registries, it is maintained that the incidence of lung cancer in never-smokers has not yet been thoroughly and comprehensively investigated [19]. This study aimed to determine the percentage of non-smokers among patients undergoing anatomical resection of the lung tissue due to lung carcinoma and to investigate the differences between the course of lung cancer, survival, and the comorbidities in the groups of patients with lung cancer depending on the history of tobacco smoking. Additionally, the study compared risk factors and outcomes between smokers and non-smokers. The term *non-smokers* refers to persons who have smoked fewer than 100 cigarettes in their lifetime, including lifetime non-smokers [20].

## 2. Materials and Methods

### 2.1. Study Design

A study was performed on a cohort of 923 patients who underwent radical anatomical resection of the lung tissue (segmentectomy, lobectomy, bilobectomy or pneumonectomy) due to lung cancer between May 2012 and December 2021. The medical records of all patients operated on due to lung cancer in our center were analyzed in detail and collected in a dedicated database. The database collected data on the previous medical history, exposure to harmful environmental factors and stimulants, family cancer history, precise assessment of the stage of cancer along with the exact diagnosis of the type of lung cancer determined on the basis of the postoperative histopathological examination and the results of perioperative care. In addition, each of the operated patients remained under the supervision of the outpatient clinic, which enabled the assessment of long-term treatment results. The day of the surgical procedure was the starting point of the observation. The observation of patient survival was conducted up to five years after the surgery. Data about the patients’ survival were collected up to 1 May 2022. All further outcomes were considered incomplete.

The inclusion criteria were: histopathologically confirmed primary adenocarcinoma or squamous cell carcinoma.

The exclusion criteria were: age < 18 years old, histopathologically confirmed adenosquamous carcinoma, secondary lung neoplasm confirmed histopathologically, occurrence of more than one histologically different tumor in post-operative material.

The detailed study design is presented in Figure 1 below.

The mean age of patients in the cohort was 66.8 +/− 7.54 years (range 38–81 years). There were 403 women (43.66%) and 520 men (56.33%). In total, 204 of the patients were non-smokers (22.11%, 95% CI: 19.43–24.79%). Further information about the group is presented in the results section. The study was approved by Bioethics Committee of Medical University of Silesia in Katowice

### 2.2. Statistical Analysis

The data was presented as the number of cases with percent value for categorical variables and mean +/− SD for quantitative variables. The normality assumption was tested for each quantitative variable based on a graphical interpretation of the Q–Q plot and histogram. The odds ratios with a 95% confidence interval were calculated for the categorical variables. To determine the differences between categorical variables, the Chi2 Pearson’s test was performed; for qualitative variables we used the *t*-test for normally distributed variables and the Mann–Whitney U test for non-normally distributed variables. The Kaplan–Meier method was used to determine the survival probability among groups. Comparison of survival was performed using the Log-rank test with Mantel correction in case of comparing more than 2 groups. To assess the influence of more than one variable on patients’ survival, Cox proportional hazard model was performed. *p* values lower than 0.05 were considered significant. Analysis was performed using the R language in Rstudio software.

## 3. Results

### 3.1. Age of Patients and Concomitant Neoplastic Diseases

The mean age of smoking patients was significantly higher than non-smokers (Table 1). No difference in the occurrence of neoplasms among family members was found.

We found that non-active smoking individuals were suffering from other neoplastic diseases compared to active smokers (Table 2).

### 3.2. Tumor Characteristics

We found no significant differences in the tumor stage, grade, vessel and pleural involvement status, nor tumor size (Table 3 and Table 4). However, adenocarcinoma was statistically a more frequent tumor among non-smokers compared to smokers.

### 3.3. Survival Analysis

For purposes of survival analysis, only patients with R0 resection status were included. No significant differences in survival rate were found among the groups of non-smokers vs. smokers (Table 5, Figure 2).

No significant differences in survival between cancer histologic subtypes were found, however we found significantly better survival among non-smoking women compared to non-smoking men (Figure 3; Table 6).

The Cox proportional hazard model including the age, gender, smoking status and histologic type showed a weak influence of age on survival rate (HR = 1.01); male gender had 1.29 HR at *p* = 0.09. The model consisting of only gender and smoking status showed a significant influence of male gender on the risk of death (Table 7).

## 4. Discussion

In our single-center study of 204 non-smokers and 719 smokers, we investigated the characteristics of lung cancer among non-smokers (LCNS) and their comparison to lung cancer among smokers (LCS) with adenocarcinoma and squamous cell carcinoma. In the studied groups, we observed differences in the age of diagnosis, the histological subtype, and the incidence of other neoplastic diseases. Our analysis revealed significantly better survival among non-smoking women compared to non-smoking men. Moreover, it should be noted that we did not observe significant differences in the survival time of the patients with LCNS compared to LCS at the specified time intervals and between the TNM scale and tumor sizes, the other examined pathomorphological parameters.

In a study of two American Cancer Society Cancer Prevention Study, cohorts of 940,000 patients who died of lung cancer with no history of smoking in 1959–2000, men showed higher mortality than women in the study group, 17.1 and 14.7 per 100,000 people per year in cohort two, respectively. In both studied cohorts, there was a significantly higher proportion of women in the group of non-smokers with lung cancer than men (382,854 vs. 94,041 in cohort I (1959–1972) and 341,643 vs. 122,563 in cohort II (1982–2000), respectively [21]). The incidence of lung cancer in non-smoking men and women was from 4.8 to 13.7 cases per 100,000 people per year and from 14.4 to 20.8 per 100,000 people per year, respectively, in the analysis of six cohorts mainly from Sweden and the USA in 1971–2002. Therefore, the obtained results supported the increased incidence of lung cancer in non-smoking women compared to non-smoking men observed in other studies [22]. Non-smoking individuals were 22.11%, 95% (19.43–24.79%) of all patients undergoing lobectomy due to primary adenocarcinoma or squamous cell carcinoma. Bade and Cruz reported in their study that 25% [1] of worldwide lung cancer cases occur among non-smokers [20]. These estimations are similar to our results. Studies report that in certain parts of the world, like Asia, non-smoking lung cancer tends to occur more frequently among women compared to men. Our study did not show any difference in the gender distribution among groups, both consisted of slightly fewer women than men.

Our analysis revealed the increased incidence and risk of neoplastic diseases in the LCNS group (OR: 1.68, 95% CI: 1.13, *p* = 0.013) compared to the LCS group. Factors that could play a key role in the formation of lung cancer in non-smokers are similar to the factors considered to be associated with the higher prevalence of the other cancers. An analysis of 183,248 patients showed a significant impact of the metabolic syndrome (MS) on the formation of various cancers and the increased risk of lung cancer (OR: 1.11, 95% CI: 1.05–1.16) and both pre-menopausal endometrial cancer and post-menopausal endometrial cancer (respectively: OR: 2.14, 95% CI: 1.74–2.65, and OR: 2.46, 95% CI: 2.20–2.74) [23]. To our knowledge, there is currently no relevant data on the comorbidity of neoplastic diseases in that group of patients. However, the occurrence of mutations and polymorphisms of genes playing an essential role in the process of carcinogenesis in the lungs and other sites, as shown in the current literature, such as EGFR, ALK, TP53, BRCA1/2, YAP1 [24,25,26], seems to make the coexistence of the other cancers in this group of patients less unexpected.

In our study, lung adenocarcinoma (LUAD) was more common than lung squamous cell carcinoma (LSCC) in non-smokers (62.25% vs. 37.75%). This is consistent with the results of most studies, where lung adenocarcinoma (LUAD) was the most common lung cancer among non-smokers [18,21,27,28,29,30]. The reason for the incidence of LUAD as the most common histological subtype in non-smokers may be the gene mutations observed more often in non-smokers with lung cancer. These mutations drive the development of this histological subtype [24,31], as well as the increased occurrence and intensity of predisposing factors to the development of adenocarcinoma in this group of patients, such as passive smoking [32,33] and obesity, which were more often observed in the group of non-smokers with LUAD compared to smokers, although the significant impact of the considered factors on the development of adenocarcinoma remains ambiguous [34,35,36]. Obtained results differ in the incidence rates of adenocarcinoma and squamous cell carcinoma within the study groups of non-smokers and smokers, which is probably due to the differences in group sizes and ethnic races between the other studies, as well as other histological cancer subtypes included in the studies [21,27,28,29,30].

The effect of lifetime smoking on lung cancer survival is ambiguous. A great number of studies have shown a significantly longer survival time from the moment of lung cancer diagnosis in non-smokers compared to smokers. A large, 13-cohort study spanning lung cancer patients from 1960 to 2004 revealed among European descent patients a significantly higher mortality rate in the group of smokers compared to the group of non-smokers. Th lung cancer death rates of lifetime smoker men were 21.9 times higher than never-smoker men, and for lifetime smoker women cancer death rates were 13.7 times higher compared to never-smoker women [37]. A study conducted in the Czech Republic in 2021 among 2439 lung cancer patients found that non-smoking patients were diagnosed with lung cancer at a later stage, but non-smokers had better survival rates than smokers [38]. Another analysis of 3380 smokers and 334 never-smokers diagnosed with lung cancer in 2003–2016 showed significantly higher overall survival among never-smokers compared to smokers; the 5-year survival rate was higher in never-smokers compared to smokers (57.9% vs. 42.6%, *p* = 0.05) [39]. However, there are also studies in which, similarly to the results of our study, no significant differences in the survival rates between the groups of smokers and non-smokers were observed [40,41].

No meaningful differences were observed in the overall survival in the 1-, 2-, and 5-year observation times in our study between the smokers and the never-smokers. However, our study groups differed from the groups in other studies due to the criteria for the inclusion and exclusion of patients from the study we had established. The differences in the observed survival times within groups of patients depending on the history of smoking in various studies and within the results of our study may be due to the differences in the number of non-smokers in the study group, the incidence, and the histological subtypes of lung cancer, the different proportions in staging in both groups of patients, and the various treatments after diagnosis. In our analysis, the percentage of non-smokers was 22.11%. We did not observe significant differences in staging and grading between the smokers vs. the non-smokers groups. In the group of non-smokers, a large proportion of the patients (58.83%) had staging I (IA2, IA3 and IB), similar to the group of smoking patients (56.33%). That makes our study different from Subramanian et al. which also showed no differences in the survival of patients from both groups, but in the study group as many as 62.5% of patients had stage III or IV [41]. Similar to the study by Nemesure et al., stage III and IV in the group of smokers and non-smokers amounted to 65.5% and 61.4%, respectively, which showed significant differences in the survival of patients with lung cancer depending on the smoking status [39]. Our study, which differed from other studies with the participation of patients in individual groups with different staging, allowed us to conclude the differences in survival of patients with a lower stage because the group we examined contained a smaller percentage of patients with staging III and IV compared to the cited studies, being 17.15% and 15.58%, respectively, in the group non-smokers and smokers. This resulted directly from the characteristics of the study group, which consisted of patients qualified for the surgical treatment of lung cancer, i.e., patients at stage IA1–IIIA, and cases of higher stages of the disease were incidental. Even though the study group contained an equal distribution of staging, the lack of differences we observed in terms of the survival of non-smokers and smokers seemed to provide new information in light of previous studies, which was particularly important in the group of patients with low staging, who were diagnosed with adenocarcinoma and squamous cell carcinoma.

It seems that non-smokers should have a longer survival time due to the expected lower number of comorbidities that increase the risk of postoperative complications and worsen the prognosis. In our study, we found improved survival among the non-smoking women compared to the men; no differences in survival between the smokers and the non-smokers were found. Apart from the survival time itself, we did not have reliable information on the patients’ further post-operative status regarding smoking status after the procedure. An important limitation of our study in terms of survival results was also the lack of information about the percentage of smokers who continued to use tobacco after the procedure. We also had limited information on passive smoking, exposure to mutagenic risk factors, and mutations in the genes involved in lung cancer in non-smokers. The lack of reliable information on the pathogenetic background of the studied subtypes of lung cancer in non-smokers, which may have an impact on survival, may also be a limitation of the results presented in the study, allowing the impact of possible heterogeneity of lung cancer patients in the group of non-smokers to effect the obtained results in terms of survival.

A large amount of data regarding lung cancer in non-smokers remain unclear, although the results obtained by various centers around the world provide a lot of valuable information. In the future, this information may contribute to a more accurate characterization of the LCNS, creating the basis for shaping future lung cancer prevention programs including a group of patients with precisely defined risk factors, and also increasing sensitivity to this group of patients in clinical practice. To our knowledge, the results of our study may contribute to future more detailed research on the observed increased incidence of comorbidity of neoplastic diseases other than lung cancer in the LCNS group patients, further indicating no significant differences in the detection of lung cancer in this group of patients compared to a group of smoking patients. The observed lack of difference in the survival rates may further illuminate the importance of the problem now posed by LCNS, ranked seventh as a stand-alone malignant disease in the incidence of malignant tumors in general, only improving in importance due to the decrease in smoking trends and the increase in the importance of factors potentially predisposing to the development of LCNS. The selected patients with adenocarcinoma and squamous cell carcinoma seemed to reflect well the characteristics of the entire group of patients with NSCLC, which is the most common group of lung cancer tumors not only in the group of non-smokers but also smokers. A significant share of non-smokers in the study group (over 20%) compared to other current studies, with a comparable median age in both groups of patients, was undoubtedly a strength of our study. Unfortunately, we did not include in our analysis some important risk factors for the development of LCNS, such as passive smoking or obesity, due to the lack of reliable information in this regard. It remains a topic that requires deeper research. As in the case of information on the impact of environmental pollution and occupational exposure on factors predisposing to the occurrence of lung cancer, these factors are difficult to measure objectively and reliably. In our study, in regard to the environmental factors, our patients were diagnosed and operated on in one center, living in an area with similar air pollution. The study group, including 923 patients with adenocarcinoma and squamous cell carcinoma, in the light of the selected 10-year period, seemed to be well selected in terms of time, due to the changing standards of oncological diagnostics and treatment over the years. The homogeneity of the study group in terms of ethnicity seemed to provide a better reference of the results to this ethnic group in which modifiable risk factors for lung cancer in non-smokers seemed to increase.

The limitation of our study was the lack of information about the further treatment of the patients. Additionally, no relevant information about passive smoking was obtained from the patients.

## 5. Conclusions

In conclusion, 22.11% of the patients undergoing radical anatomical resection of the lung tissue due to lung cancers were non-smokers. There were no significant differences in the stage of disease nor between the tumor grade among groups. In non-smokers with lung cancer, other cancers were more common. Adenocarcinoma tended to be more frequent among non-smokers than squamous cell carcinoma.

Non-smoking women had a significantly better survival rate compared to non-smoking men. No differences in survival rate were found between the smokers group compared to the non-smokers group. The differences between lung cancer patients with and without a smoking history should be further explored to better establish future lung cancer screening and prevention programs for non-smokers.

## Figures and Tables

**Figure 1 cancers-15-01896-f001:**
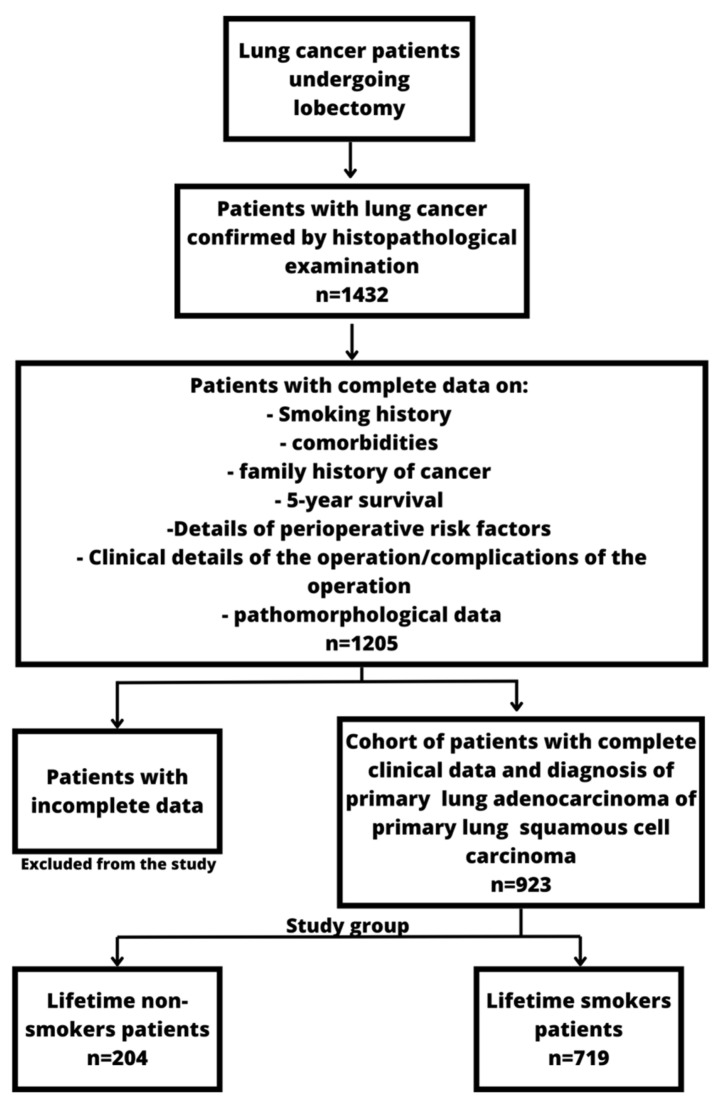
Study design flowchart.

**Figure 2 cancers-15-01896-f002:**
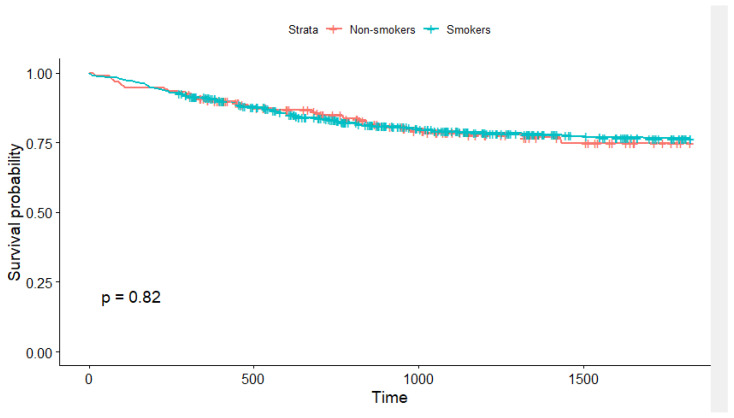
Survival curves for smokers and non-smokers, log-rank test.

**Figure 3 cancers-15-01896-f003:**
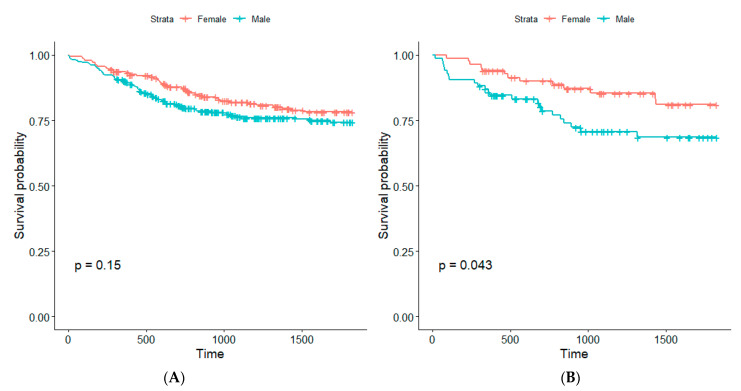
Survival analysis between gender among smokers (**A**) and non-smokers (**B**).

**Table 1 cancers-15-01896-t001:** Age of patients.

	Non-Smokers		Smokers		*p*	Cohen’s D
	Mean	sd	Mean	sd		
Age	68.0	7.95	66.0	7.41	0.043	0.164

**Table 2 cancers-15-01896-t002:** Comparison of former cancer history among smokers and non-smokers.

	Non-Smokers		Smokers	*p*	OR	95% CI		
	n	%	n	%				
Former neoplastic diseases	44	21.57%	101	14.05%	0.013	1.68	1.13	2.50
Renal cell carcinoma	2	0.98%	4	0.56%	0.864	1.77	0.32	9.73
Breast carcinoma	7	3.43%	15	2.09%	0.394	1.67	0.67	4.15
Uterine carcinoma	7	3.43%	5	0.70%	0.007	5.07	1.59	16.16
Former lung carcinoma	6	2.94%	8	1.11%	0.118	2.69	0.92	7.85
Prostate carcinoma	8	3.92%	13	1.81%	0.124	2.22	0.91	5.42
Laryngeal carcinoma	2	0.98%	6	0.83%	0.999	1.18	0.24	5.87
Carcinoma of the ovary	0	0.00%	2	0.28%	0.999	0.00	-	
Thyroid carcinoma	0	0.00%	1	0.14%	0.999	0.00	-	
Pancreatic carinoma	0	0.00%	1	0.14%	0.999	0.00	-	
Stomach carcinoma	1	0.49%	2	0.28%	0.999	1.77	0.16	19.58
Colorectal carcinoma	3	1.47%	11	1.53%	0.999	0.96	0.27	3.48
Soft tissue sarcomas	0	0.00%	1	0.14%	0.999	0.00	-	
Melanoma	0	0.00%	2	0.28%	0.999	0.00	-	
Urinary bladder carcinoma	3	1.47%	15	2.09%	0.784	0.70	0.20	2.44
Other	8	3.92%	19	2.64%	0.471	1.50	0.65	3.49
Neoplasms in family	34	16.67%	136	18.92%	0.529	0.86	0.57	1.30

**Table 3 cancers-15-01896-t003:** Comparison of tumor characteristics among smokers and non-smokers.

	Non-Smokers		Smokers		*p*
	n	%	n	%	
T parameter					
T1a	5	2.45%	23	3.20%	0.65
T1b	46	22.55%	125	17.39%	
T1c	34	16.67%	150	20.86%	
T2a	51	25.00%	171	23.78%	
T2b	21	10.29%	78	10.85%	
T3	29	14.22%	108	15.02%	
T4	18	8.82%	64	8.90%	
N parameter					
N0	165	80.88%	603	83.87%	0.3337
N1	23	11.27%	79	10.99%	
N2	16	7.84%	37	5.15%	
M parameter					
M1a	0	0.00%	1	0.14%	
Stage					
I	120	58.83%	405	56.33%	0.6314
II	49	24.02%	202	28.10%	
III	35	17.15%	111	15.44%	
IV	0	0.00%	1	0.14%	
Histologic type					
Adenocarcinoma	127	62.25%	379	52.71%	0.0194
Squamous cell carcinoma	77	37.75%	340	47.29%	
Grading					
G1	12	5.88%	47	6.54%	0.816
G2	97	47.55%	336	46.73%	
G3	75	36.76%	288	40.06%	
not given	20	9.80%	48	6.68%	
Vessels invasion					
Lymphatic invasion	33	16.18%	138	19.19%	
Vascular invasion	28	13.73%	101	14.05%	
Pleural invasion					
0	137	67.16%	510	70.93%	0.246
1	39	19.12%	131	18.22%	
2	17	8.33%	34	4.73%	
3	3	1.47%	9	1.25%	
3	188	92.16%	629	87.48%	0.119
R1	9	4.41%	62	8.62%	
R2	1	0.49%	2	0.28%	
STAS	39	19.12%	124	17.25%	0.536

**Table 4 cancers-15-01896-t004:** Tumor size comparison between groups.

Variable	Min	Max	Median	q1	q3	Mean	sd	se	*p*
Non-smoking	8	120	30	20	46.2	36.6	21.7	1.52	0.283
Smoking	5	200	30	22	50	38.4	22.8	0.849	

**Table 5 cancers-15-01896-t005:** Survival rate among groups.

	Non-Smokers			Smokers		
	Survival Rate	95% CI		Survival Rate	95% CI	
1-year survival rate	0.904	0.863	0.947	0.911	0.889	0.932
2-years survival rate	0.850	0.799	0.904	0.833	0.804	0.864
5-years survival rate	0.749	0.683	0.822	0.765	0.730	0.802

**Table 6 cancers-15-01896-t006:** Multivariate Cox proportional hazard model summary.

Variable	B	HR	95% CI		*p*
Smoking	−0.02	0.98	0.68	1.40	0.91
Squamous cell carcinoma	0.129	1.14	0.84	1.54	0.41
Age	0.03	1.03	1.01	1.05	0.00
Gender (Male)	0.30	1.35	0.98	1.85	0.06

**Table 7 cancers-15-01896-t007:** Multivariate Cox proportional hazard model excluding age.

Variable	B	HR	95% CI		*p*
Smoking	0.19	1.21	0.89	1.64	0.21
Squamous cell carcinoma	−0.07	0.93	0.65	1.33	0.69
Gender (Male)	0.35	1.42	1.03	1.94	0.04

## Data Availability

The data presented in this study are available in this article.

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
