# Peer review of "Incidence of Concomitant Neoplastic Diseases, Tumor Characteristics, and the Survival of Patients with Lung Adenocarcinoma or Squamous Cell Lung Carcinoma in Tobacco Smokers and Non-Smokers—10-Year Retrospective Single-Centre Cohort Study"

_cancers, 2023, doi:10.3390/cancers15061896_

Round 1
Reviewer 1 Report
In this article, the authors retrospectively reviewed the characteristics of lung cancer smokers and non-smokers in a single center cohort data. This manuscript is potentially acceptable for publication if the authors could address the following issues:
(1) In the abstract, the authors should state more clearly what are the subsequent investigations that should be done according to their findings.
(2) In table 3, the authors should categorize the stage as Stage I to IV and see if the stage between smokers and non-smokers remain significantly different.
(3) I noticed that the cohort contains both completely resected (R0) and non-complete resected (R1-2) lung cancer patients, and the complete resection rate was unbalanced between groups (92% vs 87%). The authors should perform survival analysis in the sub-cohort with only completely resected tumors and validate if survival remains comparable between smokers and non-smokers.
(4) The authors should perform variable-balancing statistical methods (e.g. propensity score matching) between groups (smoker vs non-smoker; male vs female) before performing survival analysis, otherwise the survival difference detected could be results from bias.
(5) An important feature in non-smoking lung cancer is the enrichment of driver mutations, the authors should report the proportion of patients harboring targetable driver mutations (EGFR, ALK, ROS1, MET, etc) and whether these patients received targeted therapies.
Author Response
Dear Reviewer,
We respond to the review below
Comments and Suggestions for Authors
In this article, the authors retrospectively reviewed the characteristics of lung cancer smokers and non-smokers in a single center cohort data. This manuscript is potentially acceptable for publication if the authors could address the following issues:
- In the abstract, the authors should state more clearly what are the subsequent investigations that should be done according to their findings.
We modified the abstract and defined more precisely what seems to be the most important in the extensive context of further research on the topic of differences between non-smokers and smokers with lung cancer.
(2) In table 3, the authors should categorize the stage as Stage I to IV and see if the stage between smokers and non-smokers remain significantly different.
Thank you for a comment, Table 3 has been corrected.
(3) I noticed that the cohort contains both completely resected (R0) and non-complete resected (R1-2) lung cancer patients, and the complete resection rate was unbalanced between groups (92% vs 87%). The authors should perform survival analysis in the sub-cohort with only completely resected tumors and validate if survival remains comparable between smokers and non-smokers.
Thank you for your suggestions, we have excluded R1-2 patients from survival analysis
(4) The authors should perform variable-balancing statistical methods (e.g. propensity score matching) between groups (smoker vs non-smoker; male vs female) before performing survival analysis, otherwise the survival difference detected could be results from bias.
In our study we performed multivariate Cox model instead of propensity score matching to adjust for confounder variables. We have excluded R1-2 status from analysis due to your suggestions.
(5) An important feature in non-smoking lung cancer is the enrichment of driver mutations, the authors should report the proportion of patients harboring targetable driver mutations (EGFR, ALK, ROS1, MET, etc) and whether these patients received targeted therapies.
Unfortunately we have no such an information. Our department started targetable mutations diagnostics in 2017, while our data were being collected much earlier.
Thanks for all comments and suggestions.
If there is anything else the Reviewer thinks we could improve, please suggest.
The authors
Reviewer 2 Report
The submitted manuscript examined the characteristics of lung cancer in smokers and nonsmokers and concluded that nonsmokers have a greater risk of preexisting neoplastic disorders, particularly uterine cancer. Unfortunately, the research design is flawed, resulting in overinterpreted conclusions.
Major:
Method and Table 2:
The authors did not have a subgrouping plan in place before they started the retrospective research, instead they performed subgrouping analysis based on pre-existing conditions available in the cohort. This research approach would result in false positive conclusions.
For example, when testing 10 subgroups repeatedly, the likelihood of hitting one false positive is 40%.
Please refer to this paper for detailed discussion:
Lagakos SW. The challenge of subgroup analyses--reporting without distorting. N Engl J Med. 2006 Apr 20;354(16):1667-9. PMID: 16625007.
Minor:
The decimal separators are not consistent in this manuscript.
Author Response
Dear Reviewer
We respond to the Review Report below
Comments and Suggestions for Authors
The submitted manuscript examined the characteristics of lung cancer in smokers and nonsmokers and concluded that nonsmokers have a greater risk of preexisting neoplastic disorders, particularly uterine cancer. Unfortunately, the research design is flawed, resulting in overinterpreted conclusions.
Major:
Method and Table 2:
The authors did not have a subgrouping plan in place before they started the retrospective research, instead they performed subgrouping analysis based on pre-existing conditions available in the cohort. This research approach would result in false positive conclusions.
For example, when testing 10 subgroups repeatedly, the likelihood of hitting one false positive is 40%.
Please refer to this paper for detailed discussion:
Lagakos SW. The challenge of subgroup analyses--reporting without distorting. N Engl J Med. 2006 Apr 20;354(16):1667-9. PMID: 16625007 .
Minor:
The decimal separators are not consistent in this manuscript.
Dear Reviewer
We have read the papers recommended in your response.
Accordingly, in order to avoid overinterpretation of the results of the analysis, we decided not to conclude on the increased incidence of uterine cancer in the group of non-smokers with lung cancer.
Therefore, in the analysis of differences in the incidence of cancers other than lung cancer, we present only a conclusion on the significantly increased incidence of former neoplastic diseases in the group of non-smokers with lung cancer compared to the group of smoking patients with lung cancer.
In addition, we improved the survival analysis in the manuscript by excluding patients with R1-2 status, due to the uneven distribution of the variable in the study subgroups (smokers vs non-smokers). Minor changes also appeared in the study of differences in the severity of the disease.
The abstract of the work has also undergone a slight change. As recommended, we also standardized the decimal separators.
Is this approach appropriate for the reviewer?
Does the reviewer see other opportunities to improve the study?
Thank you for the reviews received.
Round 2
Reviewer 2 Report
Thank you for keeping us informed. My main critique is the lack of a study plan in place prior to undertaking statistical analysis, which leads to over interpretation. The revised manuscript still has two types of decimal separators.
Author Response
Dear Reviewer
The response to the review is included in the attachment
Sincerely
The authors
